# Enhance the Immune Checkpoint Inhibitors Efficacy with Radiotherapy Induced Immunogenic Cell Death: A Comprehensive Review and Latest Developments

**DOI:** 10.3390/cancers13040678

**Published:** 2021-02-08

**Authors:** Adrien Procureur, Audrey Simonaggio, Jean-Emmanuel Bibault, Stéphane Oudard, Yann-Alexandre Vano

**Affiliations:** 1Hôpital Européen Georges Pompidou, Service d’Oncologie Médicale, Assistance Publique-Hôpitaux de Paris (AP-HP) Paris-Centre, F-75015 Paris, France; adrien.procureur.oncology@gmail.com (A.P.); audrey.simonaggio@aphp.fr (A.S.); stephane.oudard@aphp.fr (S.O.); 2Hôpital Européen Georges Pompidou, Service d’Oncologie Radiothérapie, Assistance Publique-Hôpitaux de Paris (AP-HP) Paris-Centre, F-75015 Paris, France; jean-emmanuel.bibault@aphp.fr; 3Centre de Recherche des Cordeliers, Sorbonne Université, Inserm, Université de Paris, F-75006 Paris, France

**Keywords:** radiotherapy, radioimmunotherapy, immunogenic cell death, immune checkpoint inhibitors, combination therapy, cancer treatment

## Abstract

**Simple Summary:**

Efficient antitumoral immune response is conditioned by a regulated cell death called immunogenic cell death. Many immunologic cell death inducers are currently used in cancer treatment. Among them, radiation therapy is an adaptable, well tolerated and widely used modality of treatment in modern oncology. Moreover, there is growing evidence for synergistic mechanisms between radiotherapy and immune checkpoint inhibitors. Although pre-clinical concepts are numerous, robust clinical evidence is scarce. Radioimmunology is a rapidly evolving discipline with several ongoing clinical trials. In this review, we (i) explain the rationale behind radiotherapy and immune checkpoint-inhibitor association in the light of the most recent knowledge, (ii) provide the results of the latest clinical trials evaluating radiation therapy and immune checkpoint association and (iii) explore the future directions of radioimmunology research.

**Abstract:**

The immunogenic cell death (ICD) is defined as a regulated cell death able to induce an adaptive immunity. It depends on different parameters including sufficient antigenicity, adjuvanticity and favorable microenvironment conditions. Radiation therapy (RT), a pillar of modern cancer treatment, is being used in many tumor types in curative, (neo) adjuvant, as well as metastatic settings. The anti-tumor effects of RT have been traditionally attributed to the mitotic cell death resulting from the DNA damages triggered by the release of reactive oxygen species. Recent evidence suggests that RT may also exert its anti-tumor effect by recruiting tumor-specific immunity. RT is able to induce the release of tumor antigens, to act as an immune adjuvant and thus to synergize with the anti-tumor immunity. The advent of new efficient immunotherapeutic agents, such as immune checkpoint inhibitors (ICI), in multiple tumor types sheds new light on the opportunity of combining RT and ICI. Here, we will describe the biological and radiobiological rationale of the RT-induced ICD. We will then focus on the interest to combine RT and ICI, from bench to bedside, and summarize the clinical data existing with this combination. Finally, RT technical adaptations to optimize the ICD induction will be discussed.

## 1. Introduction

Cancer is the second leading cause of death worldwide and the first in high-income countries [1]. Last decade, the development of immune-checkpoint inhibitors (ICI) has radically changed the clinical practice in multiple tumor types. For instance, Pembrolizumab, a humanized IgG4/kappa anti-PD1, is indicated as frontline therapy for advanced melanoma, for PD-L1 ≥ 50% advanced non-small cell lung carcinoma and for cisplatin-ineligible patient with advanced urothelial cancer. The Food and Drug Administration (FDA) has approved this ICI both for mismatch repair deficient tumors and for high mutational burden tumors, regardless of histology. However, apart from the above cases, most of the patients treated with ICI monotherapy will not have any benefit [2]. One of the reasons is the lack of efficient preexisting immune response.

The immune response must be initiated with a unique type of programmed cell death: the immunogenic cell death (ICD). ICD leads to the release of damage-associated molecular patterns (DAMPs) that is required for dendritic cell maturation and activation. Radiation therapy (RT) is a powerful tool to induce ICD and exhibits many advantages: its safety profile is well known; it could be easily associated with various other therapeutics and RT schedule and sequences could be optimized to improve its immune effects. Rare tumoral regression outside the radiation field had been reported for a long time now [3]. That phenomenon, better known as abscopal effect, is related to the induction of systemic CD8^+^-dependent immune response [4,5,6]. Today, the association of RT and ICI is relevant, especially because of the up-regulation of checkpoint proteins after tumor irradiation. On the other hand, like most of the ICD inducers, RT has been developed from the paradigm of maximum tolerated dose, not immune-optimized schedule [7]. Yet, it is now clear that greater cytotoxicity is not systematically associated with optimal immunogenicity [8]. The optimal regimen, dose per fraction and ICI association remain to be defined and evaluated within large cohorts.

The objective of this review is to provide an up-to-date overview of the RT-ICI association. First, we will describe the biological rational of the RT-induced ICD, and its potential synergy with ICI. Then we will summarize the main results of combination trials (from bench to bedside). Finally, we will discuss different strategies and RT technical adaptation to improve the efficacy of RT-ICI combination.

## 2. The Biology of Radiotherapy-Induced ICD

### 2.1. ICD in Oncology

ICD is defined as a form of regulated cell death that is sufficient to activate an adaptative immune response in immunocompetent syngeneic hosts [9]. How does ICD lead to efficient antitumoral-response? The induction of adaptative immune response depends on three parameters: antigenicity, adjuvanticity and microenvironment.

In physiological condition, apoptosis (and in specific settings necroptosis) are more tolerogenic than immunogenic regulated cell deaths [10]. Moreover, they fail to induce immune response in the context of central tolerance. Indeed, their antigens are expressed by thymic epithelium during negative selection of lymphocytes. While cancer cells display tumor-associated antigens (neo-antigens or overexpressed embryonic antigens) that are sufficient to lead to T-cell clonal expansion. Despite the intrinsic antigenicity of tumor cells, the immune response remains conditioned by the recruitment, the maturation, and the activation of antigen-presenting cell (APC).

Adjuvanticity refers to a spatiotemporal coordinated and highly specific release or exposure of danger signals required to fully activate APC [7]. It is mediated by the release of damage-associated molecular patterns (DAMPs) allowed by ICD. ICD is triggered by the endoplasmic reticulum stress and/or reactive oxygen species (ROS)-stress [11]. It is characterized by three hallmarks: calreticulin (CALR) exposure on the cell membrane, adenosine triphosphate (ATP) secretion and high-motility group box 1 (HMGB1) release into the extracellular space [12,13]. HSP70 and HSP90 exposure are also involved in ICD.

Lastly, the tumor microenvironment conditions influence dramatically both the priming and the effector phase of the antitumoral-response [14]. An immunosuppressive microenvironment disrupts the initiation of immune response through ICD [15].

### 2.2. Radiotherapy as an ICD Inducer

Antitumoral effects of RT is not restricted to induction of double-strand DNA breaks. Ionized radiation induces ROS-stress that leads to ICD through CARL exposition, ATP and HMGB1 release [14,16]. Ionized radiation promotes a pro-inflammatory environment through the release of chemokines and adhesion molecules leading to APC recruitment [17]. Besides, IFN type I secretion through STING-signaling pathway is essential to radiation-induce antitumor response [18]. T-cell functions and infiltration are also improved, respectively with enhanced MHC class I expression over tumor cells and with vascular remodeling [19,20].

Conversely, for a second time, these chemokines recruit myeloid-derived suppressor cells (MDSC) and regulatory T cells (Treg) that promote the immunosuppressive microenvironment [21,22]. IFNγ secretion from CD8^+^ T-cell, consecutively to RT fractionation, upregulates PD-L1, leading to T cell and NK exhaustion [23]. ROS also polarized tumor-associated macrophages to alternatively activated macrophages (M2) leading to PD-L1 upregulation and immunosuppressive cytokine secretion [24]. For most patients with metastatic cancer, focal radiotherapy as a standalone treatment fails to achieve an efficient systemic immune response, mainly because of co-inhibitory receptors upregulation and TGFβ secretion from recruited immunosuppressive cells [25]. The abscopal effect remains rare. PD-L1 seems to play a critical role in post-irradiation immunomodulation [21,26]. For example, PD-L1 is associated with radiation resistance in patient derived cell line model from head and neck cancer [27]. Accumulating evidence, both pre-clinical and clinical, indicates that ICI association with radiotherapy is promising to overcome the radio-induced immunosuppressive tumoral microenvironment [28,29].

### 2.3. Radiotherapy and ICI

Irradiated tumor microenvironment is enriched with several immunosuppressive cells; MDSC expresses PD-L1 and inhibit CD8^+^ T-cell and NK-cell activity; Treg exhibits high levels of CTLA4 and PD-L1 and compromise local immune response [22]. ICI restores not only cytotoxic activity of T-cell inside the irradiated-tumoral microenvironment, but it also reinvigorates APC activity and converts the irradiated-tumor in an effective in situ vaccine [30]. A systemic activity could be obtained. In pre-clinical models, Ipilimumab (IgG1 anti-CTLA4 antibody) induces Treg depletion and raises TCD8^+^/Treg ratio [22]. Anti-PD-1 or anti-PD-L1 restore CD8^+^ T-cell cytotoxicity and indirectly causes MDSC depletion, probably via TNF secretion [21]. RT-induced ICD, RT tumoral microenvironment reprogramming, and ICI effects are summarized in Figure 1.

Nevertheless, abscopal effect remains hardly reproducible in clinical practice. Above all because optimal RT schedule, dose rate and fractionation are not clearly defined. Until now, radiotherapy has been developed on the basis of maximum tolerated dose paradigm to obtain the best radiation dose delivery. However, maximum tolerated dose is not adapted to maximize the RT pro-immunogenic properties [7]. A paradigm shift is therefore necessary and new protocols are required.

Several clinical trials have evaluated the benefits of ICI association with RT. The results highly depend on tumoral subtypes.

## 3. RT and ICI Synergism: Emerging Clinical Evidence Related to Efficacy and Safety

Given the historical development of ICIs in melanoma and non-small cell lung carcinoma (NSCLC), most robust clinical results of the combination of ICI and radiotherapy have been reported for these two cancer subtypes. Data are also emerging for head and neck, esophageal and urologic cancers. The following is a non-exhaustive summary of main available data related to the feasibility and the efficacy of such combination, from phase I clinical trials to phase II/III clinical trials. Main prospective data related to the efficacy of ICI-RT combination are summarized in Table 1.

### 3.1. Efficacy of Combined ICI and Radiation Therapy

#### 3.1.1. Locally Advanced NSCLC (LA-NSCLC)

Focusing on NSCLC, several preclinical and clinical trials have demonstrated that ICI-RT combination could induce synergistic effect and improve survival outcome. However, the optimal RT parameters remain unknown, in particular the sequencing modes of combination therapy: sequential, induction or concurrent therapy. Dozens of trials are ongoing to find the best combination strategy [45]. Colorectal models offered some answers to this issue. Two pre-clinical studies demonstrated increased benefit with concurrent ICI and RT administration. Dovedi et al. demonstrated that concomitant but not sequential administration with fractionated radiotherapy improved survival. Mechanistically, after delivery of RT, IFNγ produced by CD8^+^ T cells was responsible for mediating PD-L1 upregulation on tumor cells [46].

Concomitant schedule: most trials assessing the efficacy of a concomitant schedule in LA-NSCLC are still ongoing. Jabbour et al. recently reported in JAMA oncology the results of a phase I trial evaluating PD-1 inhibition concurrently with definitive chemoradiotherapy for NSCLC [31]. Twenty-one patients received pembrolizumab combined with concurrent chemoradiotherapy. Promising 6 and 12 months PFS of 81% and 69.7% were observed, motivating larger prospective study. Median PFS was 18.7 months, partial response was observed in 74% of patients and overall response in 16% [31]. Spaas and Lievens recently summarized ongoing trials evaluating immunotherapy-radiotherapy combinations in NSCLC [47].

Among the largest study, we can highlight:Phase III—PACIFIC-4 (NCT03833154) evaluating durvalumab vs. placebo with Stereotactic Body Radiation Therapy (SBRT) in early-stage unresectable NSCLC (estimated enrollment: 706 patients).Phase II—KEYNOTE-799 (NCT03631784) evaluating pembrolizumab in combination with chemoradiotherapy in stage III NSCLC (enrollment: 216 patients).Phase II—NCT03663166 evaluating radiation and chemotherapy with ipilimumab followed by nivolumab for stage III unresectable NSCLC (estimated enrollment: 50 patients).

Sequential schedule: the phase III clinical trial PACIFIC compared durvalumab (at a dose of 10 mg/kg intravenously) or placebo every 2 weeks for up to 12 months as consolidation therapy in 713 stage III NSCLC patients with no progressive disease after two or more platinum-based chemoradiotherapy. The delay between completion of the last radiation dose and the first durvalumab administration was 1 to 42 days. Progression-free survival was significantly improved in the durvalumab arm with a median PFS of 16.8 months (95% CI, 13–18.1) versus 5.6 (95% CI, 4.6–7.8) with placebo (HR 0.52; 95% CI, 0.42 to 0.65; *p* < 0.001). The response rate was higher (28.4% vs. 16.0%; *p* < 0.001), and the median duration of response was longer (72.8% vs. 46.8% of the patients had an ongoing response at 18 months) in the durvalumab arm [33,34]. Updated OS data confirmed the long-term benefit with a 36-months OS rate of 55.3% versus 43.5% for Durvalumab arm [32]. Interestingly, examining the timing when durvalumab was started relative to the end of chemoradiotherapy suggests that starting ICI within 14 days after completion of chemoradiotherapy was associated with a greater PFS: HR 0.39 (95% CI, 0.26–0.58) vs. 0.63 (95% CI, 0.49–0.80).

Assessing similar strategy, Durm et al. recently reported the results of a phase II trial of consolidation pembrolizumab following chemoradiation for patients with unresectable stage III NSCLC. Both times to metastatic disease, PFS and OS were improved in comparison with historical controls with respectively median value of 30.7, 18.7 and 35.8 months [35].

#### 3.1.2. Metastatic NSCLC

Concurrent and sequential ICI/RT combinations are currently under investigation, both in metastatic and oligometastatic disease. Baulm et al. have recently reported the results of a phase II single-arm study evaluating pembrolizumab within 4 to 12 weeks after completion of locally ablative therapy for oligometastatic (≤4 metastatic sites) NSCLC. No new safety signals or reduction of quality of life was observed. Median PFS from the start of locally ablative therapy was 19.1 months (95% CI, 9.4–28.7), significantly greater than the historical median of 6.6 months (*p* = 0.005) [36]. The main ongoing phase III trials are listed in Table 2. Interestingly, several trials are evaluating a potential re-induction effect after initial response with immune therapy with RT. Patients get radiotherapy on a lesion while continuing the same ICI (NCT03406468, NCT03224871).

#### 3.1.3. Melanoma

Ipilimumab and brain radiotherapy: given the frequency of brain metastases in melanoma patients, first data related to the combination of ICI and radiotherapy concerned brain radiotherapy. Reports are mainly retrospective with a potential bias on patients’ selection. Two retrospective studies reported an increased median survival for patients receiving both ipilimumab and radiotherapy. In the Yale University study, patients receiving both Ipilimumab and RT had a median survival of 21.3 months (95% CI, 6.4–26.7) vs. 4.9 months (95% CI, 3.3–10.4) for those who did not receive ipilimumab (*p* = 0.044). In the MSKCC study, the delivery of radiotherapy during ipilimumab was also associated with an increase survival. Patients treated with RT before or during ipilimumab had better OS than those treated with RT after ipilimumab (1-year OS: 65% vs. 56% vs. 40%, *p* = 0.008) [48,49].

Ipilimumab and extra cranial radiotherapy: in a recent review, Kabiljo et al. reported two prospective trials combining external beam radiotherapy with ipilimumab in melanoma [50]. Both studies lack a control group receiving ipilimumab only but nevertheless response rate were higher compared with previous prospective cohort. In both studies, the combination was well tolerated. Radiation schedules were different. In the first one, metastases were irradiated with 6 to 8 Gy, 2 or 3 times followed by ipilimumab injections. Eighteen percent of patients had partial response and 18% had stable disease [42]. In the second, radiation therapy (determined by the treating radiation oncologist, ranged between 18 and 50 Gy in 1 to 15 fractions) was performed concomitantly with ipilimumab. Fifteen percent of patients achieved complete response with a median follow-up of 55 weeks and 15% achieved a partial response [43].

Anti PD-1 and radiotherapy: a combination of anti-PD1 therapy and brain radiation therapy was also investigated in numerous retrospective trials. A recent meta-analysis led by Strokes et al. revealed a significant benefit to combine either anti CTLA-4 or anti PD-1 therapy with radiation therapy versus radiation therapy alone [51]. Another meta-analysis led by Qian et al. revealed that PD-1 inhibition was more effective than CTLA-4 inhibition, both combined with radiation therapy [52]. In this meta-analysis, response of melanoma brain metastasis was studied according to the relative timing and the type of ICI. All 75 melanoma patients were treated with Gamma Knife to a median of 20 Gy (range: 12–24 Gy). Concurrent stereotactic surgery and immunotherapy (i.e., within 4 weeks) were associated with an improved reduction in the lesion volume at 6 months in comparison with non-concurrent therapy (−94.9% vs. −66.2%, *p* < 0.001). Similar results were observed after 1.5 and 3 months. The median reduction in the lesion volume was significantly higher with anti PD-1 than with anti CTLA4 at 6 months (−95.1% vs. −75.9%, *p* < 0.001). Similar results were observed after 1.5 and 3 months. The superiority of PD-1 inhibition has recently been confirmed in several retrospective studies [53,54,55,56].The first prospective phase I trial of nivolumab, ipilimumab and extracranial radiotherapy (30 Gy in 10 fractions or 27 Gy in 3 fractions) in patients with advanced melanoma demonstrated that RT-nivolumab-ipilimumab combination appears safe compared with historical data of nivolumab and ipilimumab alone. Randomized studies are ongoing to assess whether RT increases the efficacy of ICI [44].

#### 3.1.4. Head and Neck

ICI, nivolumab and pembrolizumab, used alone or in combination with chemotherapy, are now standard of care for recurrent or metastatic head and neck squamous cell carcinoma (HNSCC) in first- and second-line settings (CheckMate 141, KEYNOTE 012, KEYNOTE 048). Despite these therapeutic advances, response rate remains low and combination strategies to enhance ICI efficacy are under investigation.

In locoregionally advanced HNSCC, chemoradiotherapy (CRT) remains the standard of care. However, despite this multimodal therapy, locally advanced HNSCC (LA-HNSCC) recurs in many patients. Prospective studies evaluating the addition of ICI to CRT for non-metastatic advanced HNSCC are lacking. Although promising phase I studies with complete response rates up to 85.3% and 78% for HPV+ and HPV- locally advanced (LA) HNSCC treated with pembrolizumab and CRT, first results of phase II and III studies were disappointing [37]. Interim analysis of phase III Javelin 100, evaluating a regimen of avelumab plus chemoradiotherapy (CRT) followed by avelumab maintenance over placebo plus CRT and placebo maintenance for LA-HNSCC were presented during the 2020 ESMO annual meeting. No significant improvement in PFS was observed with the addition of avelumab (HR = 1.21, 95% CI: 0.93–1.57, *p* = 0.92) [38]. The GORTEC 2015-01 “PembroRad” randomized trial, led by J. Bourhis, has evaluated once-daily IMRT up to 69.96 Gy concomitant with cetuximab (400 mg/m^2^ loading dose and 250 mg/m^2^ weekly) or pembrolizumab (Pembro-RT arm: 200 mg Q3W during RT) for non-operated stage III-IVa-b and cisplatin unfit patients. Neither loco-regional control (LRC) nor PFS nor OS were improved with the anti-PD1 pembrolizumab. LCR at 15 months was 59% in cetuximab-RT arm versus 50% in pembrolizumab-RT arm (OR = 1.05 (95% CI: 0.43–2.59, *p* = 0.91). 2-years PFS and 2-years OS were respectively 40% vs. 42% (HR = 0.83, 95% CI: 0.53–1.29, *p* = 0.41) and 55% vs. 62% in the cetuximab-RT arm and in the Pembrolizumab-RT arm [39]. Other phase II and III studies are evaluating the combination of radiation therapy and immunotherapy, both in early and advanced settings [57]. The main studies are listed in Table 3.

#### 3.1.5. Other Malignancies

In esophageal cancer, chemoradiation remains the standard of care for unresectable disease with a poor prognosis. Since nearly half of these patients demonstrate PD-L1 expression, several ongoing phase I–II trials are evaluating the safety and the efficacy of chemoradiation and ICI combination, both in locally advanced, metastatic settings (NCT03377400, NCT03437200, NCT02642809) or neoadjuvant settings (NCT02735239, NCT02844075, NCT03064490).

In renal cell carcinoma (RCC), data related to the putative benefit of RT in combination with ICI are emerging. The preliminary results of the single-arm trial RADVAX RCC trial were presented at the 2020 Genitourinary Cancers Symposium. Twenty-five metastatic RCC patients had one of their metastases treated with SBRT (40–50 Gy in 5 fractions) in combination with ipilimumab and nivolumab during an induction phase followed by nivolumab alone. Fifty-six percent experienced partial response (which is higher than the expected 40%), 24% stable disease and 16% progressive disease. The 12-months PFS rate was 36% (95% CI, 0.18–0.54) which is similar to the results seen in CheckMate 214. Authors suggested that SBRT may be useful for immunologically “cold” metastatic lesions [40].

The NIVES study, a phase II multicenter trial evaluated the combination of nivolumab (240 mg every 2 weeks) with SBRT (10 Gy × 3 fractions 7 days after the first infusion of nivolumab) in advanced RCC that progressed on up to two prior systemic therapies [41]. The primary endpoint was not reached with an objective response rate of 17.4% (for an expected rate of 40%). 12-months survival rate was 73.4%. Although such combinations appeared feasible and safe, the median PFS rates were not higher than those reported previously in CheckMate 025 and CheckMate 214 with nivolumab or nivolumab-ipilimumab without SBRT. If the immediate use of RT with ICI remains uncertain, these trials support the idea that RT may be useful in a selected mRCC population made up of patients experiencing dissociated response when one tumor site starts to grow whereas the other sites decrease. It could be used to eradicate ICIs resistant clone.

### 3.2. Safety of Combined ICI and Radiation Therapy: Lessons Learned from NSCLC

In the NSCLC metastatic setting, RT is often used as a palliative treatment. With the emergence of ICI, used alone or as a combination with platinum-based chemotherapy, in front line and latter setting retrospective data related to this combination concurrently with RT have been reported in many case reports and safety appears acceptable [58]. Prospective data related, even scarce, appear reassuring and are summarized below:

#### 3.2.1. Sequential Schedule

A secondary analysis of the KEYNOTE-001 phase I trial evaluated the pulmonary toxicity of pembrolizumab after previous radiotherapy for advanced NSCLC patients. Safety profile was acceptable: three (13%) patients with previous thoracic radiotherapy developed immune-related pneumonitis compared with one (1%) of those without. No toxic death occurred [59].

Phase III clinical trial PACIFIC was also reassuring. Immune-related adverse events (IrAEs) of any grade were reported in 24.2% of patients in the durvalumab arm and 8.1% of patients in the placebo group. Grade 3–4 irAEs were respectively reported in 3.4% and 2.6% of patients. Grade 3–4 pneumonitis occurred respectively in 3.4% and 2.6%. No toxic death occurred [33,34].

#### 3.2.2. Concomitant Schedule

Safety and tolerability of ICI concurrently with chemoradiotherapy for locally advanced NSCLC were prospectively evaluated in a recent phase I trial. Twenty-one patients received pembrolizumab combined with concurrent chemoradiotherapy (weekly carboplatin and paclitaxel with 60 Gy of radiation in 2 Gy per day). Five cohorts were designed with different doses and administration schedules of pembrolizumab. No dose-limiting toxic effect (defined as grade 4 pneumonitis) was observed in the five cohorts but one patient died of G5 pneumonitis in a safety expansion cohort [31].

Safety analysis of phase II trials DETERRED (chemo-radiotherapy with or without concomitant atezolizumab followed by atezolizumab maintenance) and ETOP-NICOLAS (nivolumab concurrently to chemoradiation followed by nivolumab maintenance) report an acceptable safety profile with no increased toxicity. Reassuringly, for the first 21 enrolled patients, no Grade ≥ 3 pneumonitis was observed at the end of the 3-months post RT follow-up [60,61].

The underlying pulmonary parenchymal status may favor immune-related pneumonitis, as immune-related pneumonitis is preferentially located within tumor areas involve by tumor and/or radiation fields [62].

A recent PRISMA-compliant systematic review including 51 studies (*n* = 15,398) with 35 ICI alone and 16 ICI + RT confirmed a comparable grade 3–4 toxicity in using ICI + RT (16.3%, 95% CI 11.1–22.3%) compared to ICI alone (22.3%, 95% CI 18.1–26.9%) in CNS melanoma metastases, NSCLC and prostate cancer, regardless of cancer type [63].

## 4. Increasing the RT and ICI Synergism: Schedule, ICI Partner and Sequence

### 4.1. Optimizing the Sequence and the Choice of ICI

Is a combined better than a sequential administration? The optimal schedule actually depends on the type of ICI. The main effect of CTLA4 antibodies seems to be more related to Treg depletion via antibody-dependent cell-mediated cytotoxicity (ADCC) than CD8^+^ T-cell exhaustion from CTLA4/CD28 axis. Thus, unmodified IgG1 antibodies like ipilimumab (human IgG1/kappa) induce a greater Treg depletion in mouse model than IgG2 antibodies like tremelimumab (human IgG2/kappa) [64,65]. TGFß secretion from preexisting Treg is fundamentally involved in irradiation-induced immunosuppression and inhibits both dendritic cell maturation and CD8^+^ T-cell cytotoxicity [15,66]. Pre-clinical data suggest that prior TGFß inhibition is required to trigger a radiation-induced vaccination [66]. An administration of anti-CTLA4 before RT, in order to deplete preexisting Treg, is a wise choice to reprogram tumoral microenvironment. Young et al. have demonstrated on a murine model of colorectal cancer that anti-CTLA-4 is more efficient when it is administrated seven days before a single 20 Gy dose rather than one or seven days after [67]. This concept is also supported by the results of a retrospective study of 29 patients with advanced melanoma. The median overall survival was lower when RT was performed during ipilimumab induction phase compared to after ipilimumab induction phase [68]. Whether in preclinical or clinical settings, anti-CTLA4 and RT association efficacy is hindered by RT-induced PD-L1 upregulation [21,42]. By the way, cases of systemic effect of ipilimumab-RT association remains infrequent [69,70]. Similarly, in triple negative breast cancer mice model AT-3, Verbubrugge et al. have demonstrated that CD137 and CD40 agonist antibodies associated with RT were an effective regimen to slow tumor growth, but it was unable to cure mice. While the association of anti-PD-1, CD137 agonist and RT was the only combination capable of achieving tumor rejection. In this model, RT induces enrichment of CD137^+^ PD-1^high^ CD8^+^ T-cell that recognize AT-3 specific antigen. The efficacy of radioimmunotherapy appears to be highly dependent on the PD-1/PD-L1 axis [71].

The abscopal effect with anti-PD-1/PD-L1 and RT association is more common, but it is often difficult to distinguish a true abscopal effect from a conventional response to anti-PD-1. Most of the clinical trials do not have a control arm and mix ICI-naïve and ICI-refractory patients. Anti-PD-1 has been administered between 0 to 28 days before RT [72,73]. Unlike anti-CTLA-4 or anti-TGFß, the optimal administration schedule of anti-PD-1/PD-L1 and other checkpoints targeting CD8^+^ T-cell (like OX40 agonist) seems to be immediately following radiation therapy [52,74]. This timing corresponds to PD-L1 upregulation with radiation-increasing interferon. What is more, anti-CTLA-4 and anti-PD-1 (or other ICI) do not exhibit redundant effect. In pre-clinical mice model, genetic elimination of PD-L1 on tumor cells or addition of anti-PD-L1 restores response to RT and Ipilimumab association in various xenograft [42]. The clinical toxicity of nivolumab and ipilimumab association is significant. Moreover, sequential approach in order to minimize immune related-adverse events and optimize ipilimumab administration should be considered. Especially because a single dose of ipilimumab has demonstrated partial tumor shrinkage [75].

Many other combinations are possible, either to target multiple immune checkpoints when the doublet is inefficient and to bypass a nonfunctional immune pathway (ex., RT + anti-PD-1 + anti-TIM3 for glioblastoma in murine models) [76]. The association of RT with a CD40 agonist is particularly promising. Indeed, CD40 is a costimulatory protein involves in CD8^+^ T-cell, B-cell and macrophage activation. CD40 agonist allows activating CD8^+^ T-cell without hSTING signaling [71]. Lastly, combinations of RT with other ICI, like anti-LAG3, anti-TIGIT, anti-VISTA, anti-BTLA, are still unexplored.

Several clinical trials are testing the addition of RT to patients who experienced disease progression on prior anti-PD-1. Nevertheless, sequential treatment appears to be less effective than combination [23,46]. In our clinical experience, tumoral response is uncommon when RT is initiated once the tumor is escaping from ICI therapy. Further trials are ongoing to specify the synergism between miscellaneous ICI and RT combination strategies. Otherwise, these trials used various RT schedules and doses per fraction. In this respect, the extrapolation of results will be challenging.

### 4.2. Optimizing the Dose and Regimen of RT

Balance between radiotherapy-induced pro-inflammatory or anti-inflammatory effect highly depends on the schedule and dose per fraction. Low dose per fraction (<1 Gy) promotes anti-inflammatory effect whereas high doses (>12 Gy) triggers destruction of vasculature and ROS excess, leading to cell death, including immune cells. It also activates TREX1 exonuclease that degrades cytosolic double-stranded DNA and consequently downregulate hSTING [8]. On the other hand, higher doses mean stronger MHC-I upregulation [19].

Dewan et al. has determined that hypofractionated radioimmunotherapy with ipilimumab is superior to a single fraction [77]. Contrary to Lugade et al. that reported a significant greater inhibition of tumor growth in B16 mouse melanoma model [78]. On second thoughts, hypofractionated radiotherapy is less cytotoxic on radio-resistant tumors but it generates better adjuvanticity [77].

Schaue et al. have proposed an hypofractionated regimen with two fractions of 7.5 Gy for best ratio cross-priming/Treg cell increase, but the optimized dose remains model dependent [79]. Indeed recently, Qin Q et al. and Quéro et al. have reported respectively 3 and 4 patients suffering from heavily pretreated and anti-PD-1-naïve Hodgkin lymphoma whom all underwent durable and complete response (CR) with association of palliative normofractioned RT and anti-PD-1 (the historical CR rate is estimated at 20%) [80,81]. Both conventional radiotherapy and stereotactic ablative radiotherapy (SARB) are capable of inducing an immune response [82]. Similarly, a low dose radiotherapy has been tested to recover CAR-T cell efficacy. Herein, a single dose of 2 Gy successfully increases sensitivity of antigen-negative tumor cells to death-receptor engagement. Thereby, even in the absence of CAR/antigen interaction, CAR-T cells induce antigen-negative tumor cell apoptosis through the extrinsic pathway with the TNF-related apoptosis-inducing ligand (TRAIL) engagement [83]. At last, low dose radiotherapy improve T-cell receptor independent cytotoxicity.

Regardless of this, a large number of preclinical and clinical studies have evaluated dose per fraction between 0.5 to 30 Gy ranging from 1 to 30 fractions, sometimes with contradictory results [73,84].

### 4.3. Optimize the Irradiation Field

As previously described, the abscopal effect requires effective T cell functions. Lymphopenia is known to negatively affect outcomes in cancer patients [85]. Radiotherapy techniques and dosimetry affect the absolute lymphocyte count (ALC) and thus may impair the desired abscopal effect. Chen et al. recently reported the analysis of 3 phase I/II trials focusing on the putative relation between ALC and abscopal response. In these trials, all 153 patients underwent combined immunotherapy and RT, in metastatic setting. The post-RT absolute lymphocyte count, as a continuous variable, positively correlated with abscopal response: 34.2% of abscopal response with ALC higher than the median value vs. 3.9% with ALC lower than the median value *p* < 0.001). Similar results were reported for pre-RT ALC (30.3% vs. 7.8% respectively, *p* < 0.001). In Cox multivariate analysis, a lower post-RT ALC was associated with poorer PFS (*p* = 0.009) and OS (*p* = 0.026) [86]. Marciscano et al. reported that irradiation of the draining lymph nodes impairs adaptive immune response mainly through chemokines expression and CD8^+^ T cell trafficking and thus may be deleterious when combined with immunotherapy [87]. Considering these observations, lymph nodes sparing radiotherapy appears as an interesting strategy to preserve lymphocytes count and function and thus to synergize with immunotherapy.

### 4.4. Combination of Radiotherapy and Other Immunotherapeutic Agents

Many biological therapeutics including cytokines, vaccines, TLR agonists or innovative immune checkpoint inhibitors have been evaluated in combination with radiotherapy to improve the abscopal response. Among these therapeutics, we will focus on cytokines GM-CSF, IL2, anti TGFβ.

#### 4.4.1. GM-CSF (Granulocyte-Macrophage Colony-Stimulating Factor)

The cytokine GM-CSF acts as an immunoadjuvant by promoting the proliferation, the maturation and the migration of the dendritic cells, improving the presentation of Tumor Associated Antigens (TAAs) and thus generating an efficient T-cell response. In 2015, a proof-of-principle trial, led by Golden et al. evaluated a concurrent radiotherapy (35 Gy in 10 fractions) to one metastatic site and GM-CSF injected daily for 2 weeks (starting during the second week of radiotherapy) in patients with solid tumors receiving chemotherapy or hormone therapy. The primary objective was the rate of abscopal response. Among the 41 enrolled patients, abscopal response occurred in 11 patients. Toxicity profile was manageable and no Grade 5 adverse event occurred [88]. Liu et al. recently described similar results [89]. They evaluated abscopal effect of local radiotherapy and GM-CSF injection in patients with metastatic thoracic cancers, including lung cancers, thymic cancer, esophageal cancer, tracheal adenoid cystic carcinoma and pleural mesothelioma. On the 30 enrolled patients, 4 experienced abscopal effect (2 lung cancers and 2 thymic cancers) and 19 experienced stable disease. Combination of GM-CSF and radiotherapy appeared as a promising combination to improve abscopal effect. A phase II led by Kwek et al. evaluated the association of GM-CSF and ipilimumab in metastatic melanoma. Twenty-two patients received induction treatment with ipilimumab 10 mg/kg every 3 weeks for four doses in combination with GM-CSF 125 μg/m^2^ for 14 days beginning on the day of the ipilimumab infusion and then consolidation treatment with GM-CSF on the same schedule for 3 months without ipilimumab. The overall response rate was 32%, the median PFS was 3.5 months and the median OS was 21.1 months, suggesting that the combination may be more effective than ipilimumab monotherapy [90].

The ongoing study NCT02648282 is evaluating the association between anti PD-1 pembrolizumab, cyclophosphamide, SBRT and GVAX in patients with locally advanced pancreatic cancer. GVAX is a pancreatic cancer vaccine made of tumor cells genetically modified to release the immuno-adjuvant cytokine GM-CSF. Primary and secondary objectives are the distant metastasis-free survival and the toxicity profile. Results are warranted.

#### 4.4.2. Interleukin-2 (IL2)

Combination of RT and interleukins have been studied for a long time. A pilot study led by Seung et al. evaluated SBRT followed by high-dose of IL-2 in metastatic melanoma or renal cell carcinoma patients. Patients received 1, 2 or 3 doses of SBRT (20 Gy per fraction) followed by IL-2 (600,000 IU/kg) every 8 h for a maximum of 14 doses. Eight of the twelve enrolled patients achieved a complete or partial response (1 Cr and 7 PR), mostly melanoma patients [91].

#### 4.4.3. Anti-TGFß

Radiotherapy is known to increase TGFß activation in the irradiated tissue and to promote immunosuppressive effects, by downregulating T-cells activation and by promoting regulatory T cells and myeloid derived suppressors cells’ activation [92,93]. Rodriguez-Ruiz et al. previously demonstrated that the combination of radiotherapy with both anti PD1 and anti-CD137 was associated with favorable effects on distant nonirradiated lesions in murine models transplanted with colorectal, melanoma or breast cancer cells [6]. In 2019, using syngeneic bilateral tumor models in which only one lesion receive radiotherapy, they demonstrated that TGFß blockade enhances radiotherapy abscopal effects in combination with anti-PD-1 and anti-CD137 immunostimulatory monoclonal antibodies. The reduction of nonirradiated tumor volume was significantly higher in the RT/anti PD-1/anti-CD137/anti-TGFß group versus the RT/anti-PD-1/anti-CD137 group. This combined treatment was associated with an increase in CD8 T cells infiltrating non-irradiated lesions and with an increased Granzyme-B expression and thus putative cytotoxic effects [94].

All of the optimization described above are summarized in Figure 2.

## 5. Conclusions

ICD is a regulated death able to elicit a specific immune response through the release of DAMPs from the dying cells. Radiation therapy is a well-known ICD inducer which may be responsible for the theoretical abscopal effect. This off-target effect is a reality in pre-clinical models but is more difficult to prove in the clinic. The emergence of ICI, a new class of immunotherapeutic agents able to release the brake on the adaptive anti-tumor immune response brings new interest in exploiting radiation-induced ICD. There is a strong biological rationale to support a synergy between RT and ICI. First clinical results for combined RT-ICI are emerging: most of them are retrospective or non-randomized phase II trials and difficult to interpret. However, these data suggest that this combination is safe with no clear increase in ICI-induced toxicity with RT. First large, randomized trials evaluating RT-ICI combination are currently ongoing and first results are awaited.

We have already some data from sequential treatment with chemo-radiotherapy followed by ICI such as the PACIFIC trial with clear positive results. Nevertheless, sequential trials raise the question of a true synergistic effect between RT and ICI and an “opportunistic” effect which means that patients who tolerate well chemoradiotherapy and show no progression at the end would be the most likely to benefit from ICI “maintenance”. In the latter case, we are not evaluating the RT-ICI synergy.

Fifty percent of cancer patients will benefit from radiotherapy. The RT-ICI association is safe, it integrates quite easily into daily practice, and it can be performed in weakened patients at an acceptable clinical cost. Compared to the targeted therapy-ICI association, adverse events and costs are lower. Similarly, the chemotherapy-ICI association is still less tolerated. The CAR-T cell and ICI association is promising. However, this strategy still presents many limitations: the choice of membrane cell targets is restricted, it involves complicated logistics, its price is prohibitive, and it has a worse toxicity profile (like cytokine release syndrome or CAR-T cell-related encephalopathy syndrome) [95]. The main limit to RT-ICI association democratization remains its inconsistent efficiency and the youthfulness of the radioimmunology science.

As seen in pre-clinical studies, schedule, dose, fractionation or ICI partner could impact the combination efficacy [4]. Many open questions will remain about how to optimize RT-ICI synergy. We will need to perform randomized trials with dedicated endpoints to unlock the full potential of radiation-induced ICD combined with ICI-restored immune response.

## Figures and Tables

**Figure 1 cancers-13-00678-f001:**
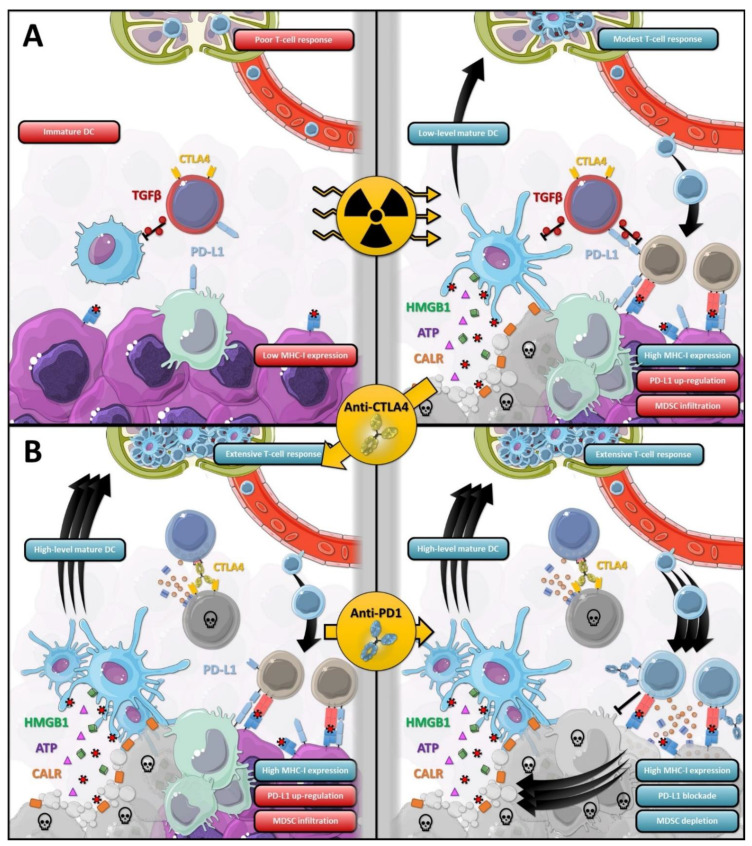
The biological consequences of RT and the synergism effects of RT-ICI association. (**A**) RT induce ICD leading to DC maturation and activation. However, DC activation is limited by Treg (red cell) through TGFβ secretion. Despite the MHC-1 upregulation, CD8^+^ T-cells remain inhibited by RT-related PD-L1 upregulation, MDSC infiltration and TGFβ secretion. (**B**) Anti-CTLA4 antibodies induce Treg depletion by ADCC, and DC activity is increased. On the other hand, CD8^+^ T-cells are still inhibited by PD-1/PD-L1 axis. It is only through the addition of anti-PD-1 or anti-PD-L1 antibodies that an effective immune response can be restored. A virtuous cycle is set up. ATP, adenosine triphosphate; CALR, calreticulin; CTLA4, cytotoxic T-lymphocyte-associated protein 4; DC, dendritic cell; HMGB1, high mobility group box 1; MDSC, myeloid-derived suppressor cell; MHC-1, major histocompatibility complex class 1; PD-L1, programmed death-ligand 1; TGFβ, transforming growth factor beta. This figure was created using Servier Medical Art templates (Creative Common Attribution 3.0), https://smart.servier.com (access on 29 December 2020).

**Figure 2 cancers-13-00678-f002:**
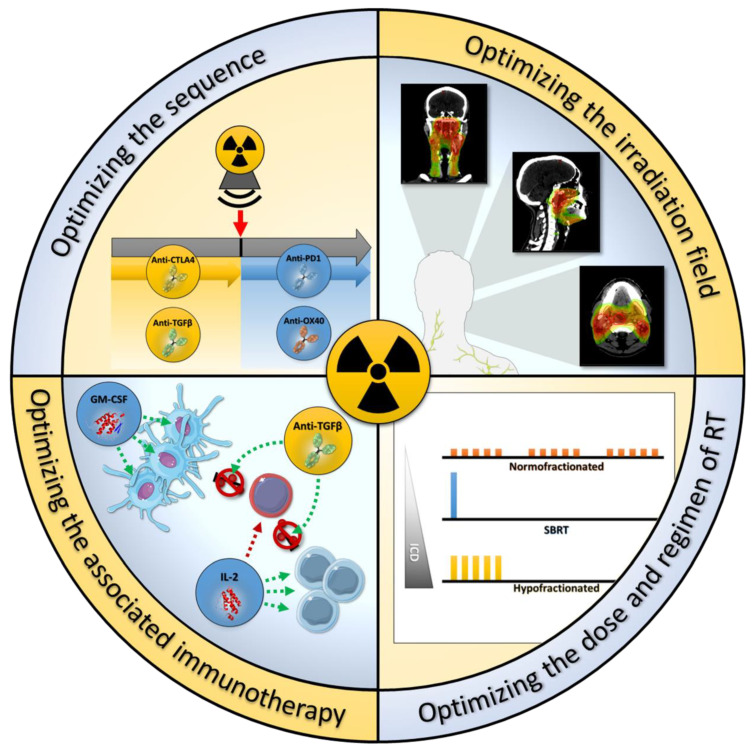
The different possible optimizations of RT to improve its immune effects. The optimisation of RT immune effects could be summarized in four categories: (i) optimizing the sequence, (ii) optimizing the associated immunotherapy, (iii) optimizing the dose per fraction and RT regimen, (iv) optimizing the irradition field. CTLA4, cytotoxic T-lymphocyte-associated protein 4; GM-CSF, granulocyte-macrophage colony-stimulating factor; IL-2, interleukin 2; PD1, programmed cell death protein 1; SBRT, stereotaxique body radiation therapy; TGFβ, transforming growth factor beta. This figure was created using Servier Medical Art templates (Creative Common Attribution 3.0), https://smart.servier.com (access on 29 December 2020).

**Table 1 cancers-13-00678-t001:** Summary of main prospective available data related to the efficacy of radiotherapy and ICI combination.

Population	Reference	Phase	Intervention	Results
Locally advanced NSCLC	[31]	I	Pembrolizumab + chemoradiotherapy	6-mo PFS rate = 81%
				12-mo PFS rate = 69.7%
				Median PFS = 18.7 mo
Locally advanced NSCLC	PACIFIC [32,33,34]	III	Durvalumab (12 mo) as consolidation therapy vs. placebo (12 mo)	ORR = 28.4% vs. 16.0% (*p* < 0.001)
				Median PFS = 16.8 mo vs. 5.6 (*p* < 0.001)
				36 months OS = 55.3% vs. 43.5%
Locally advanced NSCLC	[35]	II	Chemoradiation + pembrolizumab (12 mo) as consolidation therapy	Time to metastatic disease = 30.7 mo
				PFS = 18.7 mo
				OS = 35.8 mo
1–4 metastatic sites NSCLC	[36]	II	Pembrolizumab within 4–12 weeks after locally ablative therapy	Median PFS from the start of locally ablative therapy = 19.1 mo
Locally advanced HNSCC	[37]	I	Cisplatin-based chemoradiotherapy + pembrolizumab (concurrently + as maintenance)	CR (HPV+) = 85.3%
				CR (HPV−) = 78%
Locally advanced HNSCC	JAVELIN H&N 100 [38]	III	Avelumab + chemoradiotherapy + avelumab maintenance vs. Placebo + chemoradiotherapy + placebo maintenance	At the time of the interim analysis: no significant improvement in PFS or OS
Locally advanced HNSCC (cisplatin-unfit patients)	PembroRad [39]	II	Once-daily RT up to 69.9 Gy associated with: Cetuximab **vs**. pembrolizumab	Loco-regional-control at 15 mo = 59% vs. 50% (*p* = 0.91)
				24-mo PFS = 40% vs. 42% (*p* = 0.41)
				24-mo OS = 55% vs. 62% (*p* = 0.5)
Stage III/IV RCC	RADVAX RCC [40]	II	Nivolumab + ipilimumab + SBRT (40–50 Gy in 5 fractions)	PR = 56%
				SD = 24%
				PD = 16%
				12-mo PFS rate = 36%
2nd or 3rd line RCC	NIVES [41]	II	Nivolumab + SBRT (10 Gy × 3 fractions 7 days after the 1st infusion of nivolumab)	ORR = 17.4%
				12-mo median OS = 73.4%
Metastatic Melanoma	[42]	I	RT (6–8 Gy, 2–3 times) followed by ipilimumab injections	PR = 18%
				SD = 18%
Metastatic Melanoma	[43]	I	Ipilimumab + RT (between 18–50 Gy, in 1–15 fractions)	Clinical benefit = 50%
				PR = 15%
				CR = 15%
Metastatic Melanoma	[44]	I	Nivolumab + ipilimumab + extracranial RT (30 Gy in 10 fractions or 27 Gy in 3 fractions)	PR outside of the irradiated volume: 6/19 No progression of irradiated metastases

Abbreviations: mo: months; NSCLC: non-small cell lung carcinoma; ORR: objective response rate; CR: complete response; PR: partial response; OS: overall survival; PD: progressive disease; PFS: progression-free survival; RCC: renal cell carcinoma; RT: radiation therapy; SD: stable disease; SBRT: Stereotaxic Body Radiation Therapy.

**Table 2 cancers-13-00678-t002:** Main ongoing phase III trials with ICI/RT combinations in advanced NSCLC.

Trial Name	Study Phase	Enrollment	Stage	Experimental/Control Arm	Status
NCT03867175	Phase III	116	III	Consolidative immunotherapy with pembrolizumab +/− SBRT after first line systemic therapy	Recruiting
NCT03774732 NIRVANA-lung	Phase III	688	III/IV	Anti PD1 +/− RT 15 d after the beginning of ICI, 18 Gy	Recruiting
NCT03391869 LONESTAR	Phase III	116	IV	Local consolidation therapy: 14 d after nivolumab + ipilimumab for ICI-naive patients with metastatic NSCLC	Recruiting

**Table 3 cancers-13-00678-t003:** Main ongoing clinical trials with ICI/RT combinations in advanced HNSCC.

Trial Name	Study Phase	Enrollment	Stage	Experimental/Control Arm	Status
NCT03380394	Phase II	122	III	RT + pembrolizumab vs. RT + cisplatin	Recruiting
NCT03673735	Phase III	650	III	RT + cisplatin +/− durvalumab HPV-negative HNSCC only	Not yet recruiting
NCT02999087	Phase III	688	III	RT + cisplatin vs. RT + cetuximab vs. RT + cetuximab + avelumab	Recruiting
NCT03349710	Phase III	1046	III/IV	RT + cetuximab +/− nivolumab vs. RT + cisplatin +/− nivolumab	Recruiting
NCT03426657	Phase II	120	III	RT + durvalumab + tremelimumab	Not yet recruiting

## Data Availability

Not applicable.

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
