# Peer review of "Enhance the Immune Checkpoint Inhibitors Efficacy with Radiotherapy Induced Immunogenic Cell Death: A Comprehensive Review and Latest Developments"

_cancers, 2021, doi:10.3390/cancers13040678_

Round 1

Reviewer 1 Report

This review written by Procureur et al. systematically presents the current status of the combinatorial radiation therapy and immune checkpoint inhibitor via the immunogenic cell death mechanism. The review first introduced the mechanism underlies such strategy and the rationale to support the success of this combination. Second, the authors reviewed the current results of latest clinical trials using such combination strategy. Last, they explored future potential research directions for radioimmunology. Overall, I found this review is well written and of great potential to draw attention from wide range of researchers. I support its publication in this journal after the author consider the following suggestion.

  1. The title of this review seems to be too specialized and may compromise the readership of this work. The current title is suitable for a research article but not a review. The authors may consider to broaden the scope of this title since the contents of this review is quite comprehensive. Meanwhile, the current title "rational" should be "rationale". 
  2. The author may include discussion on the drawbacks of this strategy. In particular, compare it with other current cancer treatment strategy in terms of cost, efficacy, safety issues. Other strategy may include CAR-T cell based therapy.
  3. Page 14, line 555. Please check the use of comma. 
  4. On page 14, part 6 is patent. Does the authors want to discuss the current status of patent issues in this field or it is just a part to the required section of the journal.

Author Response

Thank you for your comments and your suggestions.

Point 1: The title of this review seems to be too specialized and may compromise the readership of this work. The current title is suitable for a research article but not a review. The authors may consider to broaden the scope of this title since the contents of this review is quite comprehensive. Meanwhile, the current title “rational” should be “rationale”.

Response 1:

  • We agree the current title may compromise the readership of our work. We suggest the following “Enhance the immune checkpoint inhibitors efficacy with radiotherapy induced immunogenic cell death: comprehensive review and latest developments”.
  • Line 2.

Point 2: The author may include discussion on the drawbacks of this strategy. In particular, compare it with other current cancer treatment strategy in terms of cost, efficacy, safety issues. Other strategy may include CAR-T cell-based therapy.

Response 2:

  • Indeed, we have not explicitly compared this strategy to other cancer therapies. On your advice, we have added a paragraph in the conclusion section summarizing the strengths and limitations of this approach compared to other cancer therapies.
  • Line 596, as follows:
    • “Fifty percent of cancer patients will benefit from radiotherapy. The RT-ICI association is safe, it integrates quite easily into daily practice, and it can be performed in weakened patients at an acceptable clinical cost. Compared to the targeted therapy-ICI association, adverse events and costs are lower. Similarly, the chemotherapy-ICI association is still less tolerated. The CAR-T cell and ICI association is promising. However, this strategy still presents many limitations: the choice of membrane cell targets is restricted, it involves complicated logistics, its price is prohibitive, and it has a worse toxicity profile (like cytokine release syndrome or CAR-T cell-related encephalopathy syndrome). The main limit to RT-ICI association democratization remains its inconsistent efficiency and the youthfulness of the radioimmunology science.”

Point 3: Page 14, line 555. Please check the use of comma.

Response 3:

  • Page 14, line 620, comma has been replaced by a point.

Point 4: On page 14, part 6 is patent. Does the authors want to discuss the current status of patent issues in this field or it is just a part to the required section of the journal.

Response 4:

  • “6. Patent” section has been suppressed (not applicable for this review).

Reviewer 2 Report

This is a well described review of literature about radiotherapy with ICI and the effects on immunogenic cell death. The link and overview of ongoing clinical trials is well done. 

Author Response

Thank you for your review and your evaluation.

Reviewer 3 Report

In this manuscript, the authors provide a review of the combination of radiotherapy and immune checkpoint inhibitors. This is an emerging field, there are numerous ongoing trials investigating checkpoint inhibitors and radiation therapy.

The topic is of utmost importance.

Major comments

1. Introduction and 2. The biology of radiotherapy-induced ICD

The sheer number of abbreviations makes it very difficult to read. Given the fact that this review addresses mainly interested oncologists, the authors should use the abbreviations with caution, or alternatively provide a list of all abbriviations used.

3. RT and ICI synergism

This part is very confusing. The authors should summerize the results of published studies, structured according to cancer subtypes, and provide the ongoing studies in a table.

4. Increasing the RT and ICI synergism

Although the authors describe optimizing the choice of immune checkpoint inhibitors, only the effects of ipilinumab are described in detail. Other checkpoint inhibitors are not mentioned in this section, just referred to as „anti-PD-1/PD1“. When discussing the choice of the ICI, the authors should not just name one drug, but describe the mechanisms of action as well as advantages and disadvantages of other often prescribed drugs as well.

Author Response

Point 5: Parts 1. Introduction and 2. The biology of radiotherapy-induced ICD; the sheer number of abbreviations makes it very difficult to read. Given the fact that this review addresses mainly interested oncologists, the authors should use the abbreviations with caution, or alternatively provide a list of all abbriviations used.

Response 5:

  • Thank you for your suggestion which enables us to improve the quality of our review. Excessive abbreviations have been removed: regulated cell death (RCD), endoplasmic reticulum (ER), ionized radiation (IR), maximum tolerated dose (MTD) and tumoral microenvironment (TME)
  • We also added abbreviation table at the end of the article (line 571).

Point 6: Part 3. RT and ICI synergism; this part is very confusing. The authors should summerize the results of published studies, structured according to cancer subtypes, and provide the ongoing studies in a table.

Response 6:

  • To clarify this section, we added a summary table of prospective data (Table 1). Line 163.
  • Main ongoing studies are summarized in other tables.

Point 7: Part 4. Increasing the RT and ICI synergism; athough the authors describe optimizing the choice of immune checkpoint inhibitors, only the effects of ipilinumab are described in detail. Other checkpoint inhibitors are not mentioned in this section, just referred to as “anti-PD-1/PD1”. When discussing the choice of the ICI, the authors should not just name one drug, but describe the mechanisms of action as well as advantages and disadvantages of other often prescribed drugs as well.

Response 7:

  • The importance of schedule administration of immune checkpoint inhibitors is mainly described for Ipilimumab. However, we didn’t wish to restrict our discussion to ipilimumab, especially since other anti-CTLA4 IgG1 are under development (like zalfrelimab/AGEN1884 or CA3054928A1). This example was intended to note the importance of a fine understanding of immune mechanisms, in order to optimize clinical trial designs. We agree that it is not desirable to name only one drug. We have therefore reworded the sentence by replacing “ipilimumab” with “anti-CTLA4 antibodies” line 367. The differences between ipilimumab and tremelimumab were described more precisely in a dedicated sentence (line 369).

Reviewer 4 Report

The current review article entitled,” Radiation therapy-induced immunogenic cell death to enhance the immune checkpoint inhibitors efficacy in cancer treatment: biological rational and latest developments” elaborated the advancement in current immunoradiation treatment modality for cancer.  Radiotherapy is one of the cornerstone treatments for cancer. It is widely known that irradiation to cancer tumor enhances immune response and a combination of RT with anti-PD1 and anti-PD-L1 inhibitors have shown added advantage in the prevention of cancer progression. The current review article well-articulated the significance of immune-radiation therapy in cancer and described development in adjuvant immuo-radiation therapy. I have a few suggestions below.

  • Authors need to provide a list of abbreviations.
  • Line 119; provides the full name of LT.
  • 2.2; Redox is known to be a major player of the innate and adaptive immune response. The author needs to highlight novel redox-sensitive components related to immune cell regulation. (PMC6410837).
  • It has been reported by a few studies that radiation-resistant cell lines exhibit increased expression of PD-L1 in HNSCC (PMC5457365). Authors may include some of the findings related to radiation resistance and highlight the significance of the combination of immuno- radiation therapy.  
  • Authors can depict the gradual development of combination therapy and list with outcome i.e. overall survival, PFS etc. in a figure/or, table.
  • Line 221; Write months(?) after 4.5
  • Line 327; irAEs.
  • Line 545; Need to be removed.

Author Response

Thank you for your comments and your suggestions:

Point 8: Authors need to provide a list of abbreviations. Line 119; provides the full name of LT.

Response 8:

  • We have added abbreviation table at the end of the article.
  • “CD8+ T-cell” has replaced “LT”.

Point 9: Part 2.2; Redox is known to be a major player of the innate and adaptive immune response. The author needs to highlight novel redox-sensitive components related to immune cell regulation. (PMC6410837).

Response 9:

  • We have added a sentence specifying the impact of ROS on the polarization of TAMs and on local immunosuppression.
  • Line 112.

Point 10: It has been reported by a few studies that radiation-resistant cell lines exhibit increased expression of PD-L1 in HNSCC (PMC5457365). Authors may include some of the findings related to radiation resistance and highlight the significance of the combination of immuno-radiation therapy.

Response 10:

  • Indeed, PD-L1 expression is a major pathway of radiotherapy resistance and it is the rationale of anti-PD1/RT association. We have included some of the findings related to radiation resistance. Including the article in HNSCC (PMC5457365).
  • Line 119.

Point 11: Authors can depict the gradual development of combination therapy and list with outcome i.e. overall survival, PFS etc. in a figure/or, table.

Response 11:

  • We summarized the main results of RT-ICI combinations from prospective trials in Table 1 cross multiple cancer types.
  • Line 163

Point 12: Line 221: Write months(?) after 4.5

Response 12:

  • “Months” has been added. Line 237.

Point 13: Line 327; irAEs.

Response 13:

  • “immune-related adverse events” has replaced “irAEs”. Line 343.

Point 14: Line 545; Need to be removed.

Response 14:

  • Sentence “6. Patent” section has been suppressed.